# *I267L* Is Neither the Virulence- Nor the Replication-Related Gene of African Swine Fever Virus and Its Deletant Is an Ideal Fluorescent-Tagged Virulence Strain

**DOI:** 10.3390/v14010053

**Published:** 2021-12-29

**Authors:** Yanyan Zhang, Junnan Ke, Jingyuan Zhang, Huixian Yue, Teng Chen, Qian Li, Xintao Zhou, Yu Qi, Rongnian Zhu, Shuchao Wang, Faming Miao, Shoufeng Zhang, Nan Li, Lijuan Mi, Jinjin Yang, Jinmei Yang, Xun Han, Lidong Wang, Ying Li, Rongliang Hu

**Affiliations:** 1Changchun Veterinary Research Institute, Chinese Academy of Agricultural Sciences, Changchun 130122, China; 18843184112@163.com (Y.Z.); zjyuanff27@163.com (J.Z.); 18380384852@163.com (H.Y.); ctcx1991@163.com (T.C.); liqian2714@yeah.net (Q.L.); zhouxtao@foxmail.com (X.Z.); qiyu0204@163.com (Y.Q.); 18624339761@163.com (R.Z.); wsc1026@126.com (S.W.); miaofaming81@163.com (F.M.); zhangshoufeng@hotmail.com (S.Z.); linan226@126.com (N.L.); mlj84321@163.com (L.M.); charminggirlyjm@163.com (J.Y.); hx19871111@126.com (X.H.); 15164390097@163.com (L.W.); 2College of Animal Science and Technology, College of Veterinary Medicine, Jilin Agricultural University, Changchun 130118, China; kejunnan0125@163.com (J.K.); jjyang2021@163.com (J.Y.)

**Keywords:** African swine fever virus (ASFV), *I267L*, deletion, virulence, replication

## Abstract

African swine fever virus (ASFV) is the causative agent of African swine fever (ASF) which reaches up to 100% case fatality in domestic pigs and wild boar and causes significant economic losses in the swine industry. Lack of knowledge of the function of ASFV genes is a serious impediment to the development of the safe and effective vaccine. Herein, *I267L* was identified as a relative conserved gene and an early expressed gene. A recombinant virus (SY18ΔI267L) with *I267L* gene deletion was produced by replacing *I267L* of the virulent ASFV SY18 with enhanced green fluorescent protein (EGFP) cassette. The replication kinetics of SY18ΔI267L is similar to that of the parental isolate in vitro. Moreover, the doses of 10^2.0^ TCID_50_ (*n* = 5) and 10^5.0^ TCID_50_ (*n* = 5) SY18ΔI267L caused virulent phenotype, severe clinical signs, viremia, high viral load, and mortality in domestic pigs inoculated intramuscularly as the virulent parental virus strain. Therefore, the deletion of *I267L* does not affect the replication or the virulence of ASFV. Utilizing the fluorescent-tagged virulence deletant can be easy to gain a visual result in related research such as the inactivation effect of some drugs, disinfectants, extracts, etc. on ASFV.

## 1. Introduction

African swine fever (ASF) is a highly contagious and severe viral disease that infects domestic and wild pigs, which has caused devastating economic losses to the swine industry worldwide. The causative agent, African swine fever virus (ASFV), has a large double stranded DNA genome ranging from 170 to 190 kilobase pairs [1]. At present, 24 genotypes have been identified based on variation of C-terminal sequence of *B646L* genes (coding p72 protein) among different isolates [2,3]. The continuous epidemic of ASF will lead to constant genomic changes of ASFV. Differential virulence strains including high- [4,5,6], low- [7,8,9,10], and non-virulence strains [11] have been isolated from naturally occurring and cell lines-adapted strains.

ASF occurred for the first time in Kenya in 1909 [12] and was first identified and reported by Montgomery in 1921 [13]. The genotype I of ASFV mainly emerged in African and part of European countries from 1921 to the mid-1990s. European countries of ASFV outbreaks, except Sardinia, eradicated this disease until the late 1990s. The genotype II of ASFV emerged in Georgia in 2007, and it soon spread to Russia and more European countries. In August 2018, ASF was first reported in China. Since then, the spread of ASF in Asia has begun. At present, 15 countries, including Mongolia, Vietnam, Cambodia, etc. have successively reported the ASF epidemic. At present, ASF outbreaks are in the Africa, Europe, Asia, North America (Dominican Republic and Haiti), and Oceania (Papua New Guinea) region currently. There are more than 60 countries and territories that have reported the disease [14,15,16]. The continued extension of ASF has caused a serious threat to the countries without ASF.

In most cases, vaccines are the first choice for disease prevention. However, ASF vaccine researches based on inactivated virus [17], naturally attenuated virus [18], DNA plasmids, proteins DNAs/antigens [19,20,21,22], live viral vector [23,24,25,26,27,28,29,30], and deletion of virulence gene [31,32,33,34,35,36,37,38,39,40] are all still in the experimental stage. The inactivated viruses and naturally-occurring attenuated viruses had been verified either ineffective or causing strong adverse reactions [41]. DNA- and protein-based subunit vaccines showed a partial protection [19,20,21]. Currently, ASF vaccine research concentrates on genetically modified live attenuated ASFV and virus-vectored subunit vaccines. Researchers have reported gene deletions, such as *9GL* [32,42], *9GL*/*UK* [43], *MGF505*/*MGF360*/*CD2v* [37], *DP148R* [33], *I177L* [38], *L7-L11L* [39], *A137R* [40], and *I226R* [44], which can attenuate ASFV virulence and protect inoculated pigs against the challenge of homologous or virulent parental isolate. On the other side, a subunit vaccine, comprised of the ASFV genes and vectored by replication-deficient human adenovirus 5 (rAd) and modified vaccinia Ankara (MVA), led to 100% protection against a fatal ASFV [23]. Noteworthily, deleting the same gene from different virulent isolates may appear different attenuation of virulence, such as *9GL* in Georgia and Malawi Lil-20/1. Additionally, doses and routes of immunization may induce differential protection effect [45].

The *I267L* gene locates at the right end of ASFV genome and consists of 267 amino acids which encodes a 30.9-*Kilodalton* protein. No significant similarity has been found among the known genes in the genetic database with *I267L*. Here, we reported that the sequence of *I267L* gene is conserved among ASFV isolates and the transcription of *I267L* gene occurs in early stage of the infection. We studied the function of *I267L* gene by constructing a recombinant ASFV (SY18∆I267L) deleting *I267L* gene and evaluated the virulence of SY18∆I267L on domestic pigs inoculated intramuscularly with 10^2.0^TCID_50_ and 10^5.0^TCID_50_. All animals developed similar fever, clinical presentation, and viremia with the same dose of the virulent parental ASFV SY18. The result demonstrated that the deletion of *I267L* gene from ASFV SY18 isolate does not influence the replication in vitro or the virulence of the parental virus in vivo.

## 2. Materials and Methods

### 2.1. Viruses and Cells

The virulent ASFV SY18 was isolated from the spleen of a domestic pig infected with ASFV in July 2018 in China. Bone Marrow-Derived Macrophages (BMDMs) were prepared from 2 to 3-month-old Landrace piglets. Briefly, the piglet was euthanized using pentobarbital. The ribs and leg bones were used to separate BMDMs and the red blood cells were lysed with Red Blood Cell Lysis Buffer (BOSTER, Pleasanton, CA, USA). The BMDMs were cultured in RPMI 1640 medium containing 10% heat-inactivated fetal bovine serum (Gibco, Waltham, MA, USA) and 10 ng·mL^−1^ granulocyte-macrophage colony-stimulating factor (GM-CSF) (*E*. *coli*-derived porcine GM-CSF protein prepared by our lab) for 7–10 days to stimulate the differentiation of BMDMs and make them infectious.

### 2.2. Homology Analysis of I267L Gene among ASFV Isolates

The amino acid sequences of *I267L* gene from different genotype isolates (GenBank accession numbers: NC_044957.1, NC_044943.1, NC_044941.1, NC_044958.1, NC_044956.1, U18466.2, LS478113.1, MH766894.2, MK333180.1, NC_044959.2, AY261365.1, AY261364.1, AY261361.1, MH025916.1, AY261360.1, and AY261363.1) were downloaded from NCBI (https://pubmed.ncbi.nlm.nih.gov/, accessed on 27 December 2021). The multiple alignment of the amino acid sequences was performed using MAFFT online website [46] (https://www.ebi.ac.uk/Tools/msa/mafft/, accessed on 27 December 2021) and Jalview 2.11.1.5 software (http://www.jalview.org/, accessed on 27 December 2021) to value the conservation of *I267L*.

### 2.3. Characteristic of the I267L Gene Expression

The BMDMs prepared in 12 well-plates were infected with ASFV SY18 at 3 MOI and the mock infected BMDMs were used as a control. The cultures were collected at 2, 4, 6, 8, 10, 12, 15, 18, 21, 24 hpi and all-time point has three repeats. The total RNA was extracted from the BMDMs using the RNAsimple Total RNA Kit (Qiagen, Dusseldorf, Germany). The elimination of ASFV genomic DNA and the transcription of RNA in vitro were completed following the instructions of PrimeScript™ RT reagent Kit with gDNA Eraser (Takara, Tokyo, Japan). The cDNA was amplified by the primes of *B646L*, *CP204L*, *I267L*, and *GAPDH* genes based on SYBR Green I-based quantitative PCR. The results were analyzed by the GraphPad Prism 8.0.2 software (https://www.graphpad.com/, accessed on 27 December 2021). The primers were shown in Table 1.

### 2.4. Construction of the SY18ΔI267L

To investigate the role of *I267L* in virulent ASFV SY18, a recombinant virus with deletion of the *I267L* gene was constructed by homologous recombination. The recombinant plasmid contains the two flanking sequences of the *I267L* and EGFP gene. The left arm of the *I267L* gene was located at 168,410–169,522 bp and the right arm was located at 170,483–171,645 bp in the ASFV genome. The expression of the EGFP gene was controlled by the p72 promoter (Pp72) [47] cloned from ASFV genome and SV40 polyA, which was located between the left and right arms of *I267L* gene. The SY18∆I267L was constructed according to the recombination between SY18 genome and the recombinant plasmid. Briefly, the recombinant plasmid was transfected into the BMDMs using jetPEI^®^-Macrophage DNA Transfection (Polyplus, Strasbourg, France). After 4 h, the BMDMs were further infected with ASFV SY18 at 3 MOI. The fluorescent cells were presented about 12 h post infection (hpi), which were screened under the Fluorescence microscope (Olmpus-IX73, Tokyo, Japan) and further purified in BMDMs through limiting dilution. The purity of SY18ΔI267L was detected by PCR. The forward primer (5′-CGTATATCTTGTGATAATGG-3′) and the reverse primer (5′-GGACTACATCTCTTCAAGCA-3′) were designed in the *I267L* gene and the right flanking region of *I267L* gene by Primer Premier 6 software, respectively. A 364 bp fragment could be amplified if the parental SY18 exists.

### 2.5. Growth Characteristic of SY18ΔI267L In Vitro

To evaluate the effect on replication while deleting the *I267L* gene from ASFV SY18, BMDMs were infected with SY18ΔI267L and ASFV SY18 at 0.01 MOI, respectively. Then the cultures were collected at 2, 12, 24, 36, 48, 72, and 96 hpi and repeatedly frozen–thawed between liquid nitrogen and water. In order to reduce the error, all times points were repeated thrice. The virus titer at every time point was tested according to a tenfold serial dilution and add 100 μL/well for each dilution to 8 wells of 96-well plate. The number of fluorescence wells for each dilution were calculated according to the Reed-Muench method for Tissue Culture Infectious Dose 50 (TCID_50_). After 72 hpi, the BMDMs infected with SY18ΔI267L and ASFV SY18 were stained by FITC-labeled p30 monoclonal antibody (diluted 1:500 in PBS) prepared by our lab. The number of the well staining fluorescence were calculated by Reed-Muench method. The cells infected with SY18∆I267L can also be directly observed under the fluorescence microscope.

### 2.6. Animal Experiments

To evaluate the effect of deleting the *I267L* gene on ASFV SY18 virulence, fifteen Landrace Pigs weighting about 20 kg were divided randomly into three groups. The pigs in the first two groups were intramuscularly (I.M.) inoculated with 10^2^.^0^ TCID_50_ or 10^5.0^ TCID_50_ of SY18ΔI267L, respectively. The pigs in the last group were inoculated with 10^2.0^ TCID_50_ of ASFV SY18 as a control. After 28 days, all pigs survived the inoculation will be challenged by intramuscular injection with 10^2.0^ TCID_50_ SY18. Clinical symptoms such as high fever, inappetence, depression, diarrhea, waddling, reluctance to stand, skin cyanosis, and arthrocele were observed and recorded daily throughout the experiment. The whole peripheral blood was collected in EDTA-containing tubes and the serum was isolated from normal blood every two days post inoculation. The pigs showing severe clinical signs were euthanized in extremis using pentobarbital and the tissues, including submandibular lymph node, tonsil, heart, lung, thymus, marrow, liver, spleen, kidney, stomach, colon, jejunum, bladder, inguinal lymph nodes, joint fluid, and muscle, were assessed the development of viral load. All samples were detected in CFX96TM Real-Time System (Thermo, Waltham, MA, USA).

### 2.7. Detection of ASFV Genome in Blood and Tissues

A probe-based real-time quantitative PCR (qPCR) targeting the ASFV *B646L* gene was performed to quantify the ASFV genomic DNA copy in the blood and tissue samples. The primer synthesis and the reaction condition were recommended by the World Organization for Animal Health (OIE) [48]. According to the standard plasmid, the standard curve is y = −3.34x + 40.1. The value of y represents the value of Ct and the value of x represents the value of Log_10_ copies/μL. Additionally, the sensitivity is about 50 copies via detecting the standard plasmid using the OIE recommended method. To establish a standard curve for absolute quantification by qPCR, the *B646L* gene was cloned into the pMD18-T vector and used as a standard plasmid. The standard curve was synthesized according to detecting the 10-fold gradient p72 plasmid. The blood and tissue samples were processed by the following methods. The tissues added PBS were ground in an automatic sample grinding machine. The blood and the supernatant of tissues were lysed by a lysis buffer (prepared by our lab) in volume ratio of 1:1. The mixture was oscillated for 5 s, boiled for 5 min, and centrifuged for 1 min. The supernatant was used as the template and was detected in CFX96TM Real-Time System.

### 2.8. Detection of Anti-p54 Antibodies

The level of antibodies against ASFV-specific protein p54 antibodies in serum was measured using an indirect ELISA (developed by our lab). The detailed process was described in the previous article [39]. Briefly, the ELISA plates (Corning, New York, NY, USA) were coated with purified p54 protein (1 μg·mL^−1^) which was expressed in prokaryotic expression system and was blocked with 5% skimmed milk. The serum sample (S) and positive control (P) were added to the ELISA plates and incubated for 1 h at room temperature (RT). The horseradish peroxidase (HRP)-labeled sheep anti-pig IgG (CWBIO, Haimen, China) used for second antibody was incubated for 1 h at RT. The chromogenic reaction began with the addition of 3,3′,5,5′-Tetramethylbenzidine (TMB) substrate (SeraCare, Delaware, USA) and ended with 2 M sulfuric acid (BEIJING SHIJI, Beijing, China). The optical density (OD) values at 450 nm were read by iMark^TM^ Microplate Reader (BIO-RAD, Hercules, CA, USA). The ratio of S/P above 0.25 is recognized as a positive sample.

### 2.9. Statistical Analysis

Statistical significance was determined using the Holm–Sidak test. A *p*-value < 0.05 was considered statistically significant and a *p*-value ≥ 0.05 was considered statistically non-significant. Similar results were obtained from three independent experiments. Statistically significant differences between groups were analyzed using GraphPad Prism 8.0.2 software (https://www.graphpad.com/, accessed on 27 December 2021).

## 3. Results

### 3.1. A Relative Conserved I267L Gene

The amino acid sequences of different isolates were analyzed using a MAFFT online website and Jalview software. The results showed that the *I267L* gene encoded 267 amino acids in most of the ASFV strains. There are several ASFV isolates—R7, R8, R25, R35, and N10—encoding 240 amino acids due to a single-base mutation, which caused an early termination of the ORF (some sequences are not shown except R8) in Figure 1. The truncation of the 27 amino acids in 3′ end sequence did not affect virus virulence [49]. The *I267L* ORFs between genotype I and II are relatively conserved. Mutations mostly occur in virus isolates among type VIII, IX, and X. The homology of the pI267L amino acids is 88.4–100%, which is relatively conserved.

### 3.2. Transcription of I267L Occurs at the Early Stage of Infection

Total RNA of BMDMs infected with ASFV SY18 or mock infection were detected by the late viral gene *B646L* (p72), the early viral gene *CP204L* (p30), *I267L* and *GAPDH* (housekeeping gene). The results showed that the *CP204L* gene maintained high expression from 2 to 24 h post infection, and the expression of the *B646L* gene kept rising from 2 to 24 h during infection in Figure 2. The *I267L* gene presented a rapid expression at the first 6 hpi and then maintained high expression. The expressing trend of *I267L* gene was similar to the *CP204L* gene. Therefore, we speculate that the *I267L* gene is an early transcription gene, and this result is consistent with the results of Cackett’s ASFV transcriptome data [50].

### 3.3. Generation of SY18 ∆I267L

The illustrations of the recombinant plasmid can be seen in Figure 3a. The diagram illuminated the design of ASFV SY18ΔI267L and the position of the *I267L* gene in ASFV SY18 in Figure 3b. The purified SY18∆I267L expressed the green fluorescence in Figure 3c. SY18ΔI267L displayed an identical growth kinetic comparing to that of the parental virus in Figure 3d. There was no significant difference (*p* > 0.5) in replication in vitro between SY18∆I267L and ASFV SY18 from the beginning to the end of the infection. Therefore, the *I267L* is not a replication-related gene for ASFV.

### 3.4. Animal Experiments

There were no pigs that survived the inoculation of 10^2^.^0^ TCID_50_/mL SY18ΔI267L, 10^5.0^ TCID_50_/mL SY18ΔI267L, and 10^2.0^ TCID_50_/mL ASFV SY18 (Table 2). All pigs inoculated with ASFV SY18ΔI267L presented ASF-associated clinical signs.

The animals inoculated with 10^5.0^ TCID_50_ SY18ΔI267L showed early fever, viremia, clinical signs, and death comparing with the low dose of SY18ΔI267L and ASFV SY18 (Figure 4a–c). The animals inoculated with 10^2.0^ TCID_50_ SY18ΔI267L presented delay fever, viremia, clinical signs, and death comparing with the same dose of ASFV SY18.

There were several pigs that produced anti-p54 antibody in the three groups in Figure 4d. The positive seroconversion rates were 4/5, 2/5, and 3/5 in the groups of 10^2.0^ TCID_50_ SY18ΔI267L, 10^5.0^ TCID_50_ SY18ΔI267L, and 10^2.0^ TCID_50_ ASFV SY18, respectively, which occurred on the 8th day post inoculation. The time of positive seroconversion is similar with our previous study on a gene-deleted attenuated SY18ΔMGF/ΔCD2v [51] and SY18ΔI226R [44].

The viral load was detected in multiple tissues of the pigs. Almost all tissues had a high viral load (up to 10^8.0^ copies/g) in Figure 4e. The tissues, including spleen, joint fluid, bladder, jejunum, and colon, had a high detection rate (15/15) in the three groups.

## 4. Discussion

ASFV is spreading in more and more countries. Strict measures, including disinfection of pigpen, surveillance of pathogen, quarantine, and cull of infected and in-contact animals have been applied to prevent and control of ASF, which cost a lot of manpower, material, and financial resources. People urgently need a safe and effective vaccine. ASFV encodes more than 150 ORFs. However, the function of about half the genes is still unknown. A comprehensive understanding of the ASFV gene will help the development of vaccines.

*I267L* is a gene of unknown function and there are no known genes or proteins to match up with it in gene and protein databases. The homology of *I267L* amino acid residues is 88.4–100%, which is relatively conserved among ASFV isolates of the different serotypes. Mazloum et al. identified *I267L* as possible genetic markers to discriminate between the closely related genotype II viruses based on the identification of SNPs or Indels within ASFV genomes [52]. In several ASFV isolates, the translation of *I267L* was terminated prematurely due to nucleotide mutations, leaving 27 amino acids untranslated, but the mutation does not influence the virulence of these isolates. The result indicates that the 27 amino acids have nothing to do with ASFV virulence. Currently, there is no research on the role of *I267L* in the process of ASFV infection. The transcription of *I267L* gene starts at an early stage and maintains a high level during the infection process.

In vitro, the deletion of *I267L* of virulent SY18 did not lead to the replication deficiency. In vivo, the animal inoculated with the same dose of SY18ΔI267L and ASFV SY18 developed the similar clinical result. The results demonstrate that *I267L* is not replication- or virulence-related gene for ASFV. There are parts of the animal that developed positive seroconversion at 8th day post infection in the three groups, which is similar to our previous research on gene-deletion attenuated SY18∆MGF/∆CD2v and SY18∆I226R. Although neither ASFV-resistant swine serum nor attenuated strain immunized pig serum can neutralize ASFV, the positive outcome of the serum is positively correlated with whether it can survive the challenge of virulent ASFV (the data were not presented). We detected the high level ASFV genome in almost all the tissues. The viral load of the bladder and colon has the high copy and detection rate. The results demonstrate that the animal mainly excretes the virus from urine and feces. This is one of the main reasons that ASFV spread rapidly and is hard to control.

The new deletant has similar characteristic in vivo and vitro with virulent ASFV SY18, which has been considered as a marker ASFV. Borac et al. developed a fluorescent ASFV strain that replaced the genes of *MGF360-13L* and *MGF360-14L* and retained the ability to cause disease in swine. Besides, the deletants, such as deletion of *X69R* [53] and *C962R* [36], also have the potential to be utilized in related research. The modified ASFV is a suitable tool to research on pathogenesis, virus-macrophage interaction, and viral antigen-based assays [54].

In summary, *I267L* is expressed at an early stage during the infection and encodes a relative conserved protein. The absence of *I267L* does not affect the replication of ASFV on primary swine macrophage in vitro. The domestic pigs could not survive the I.M. challenge with high or low doses of SY18∆I267L.

## Figures and Tables

**Figure 1 viruses-14-00053-f001:**
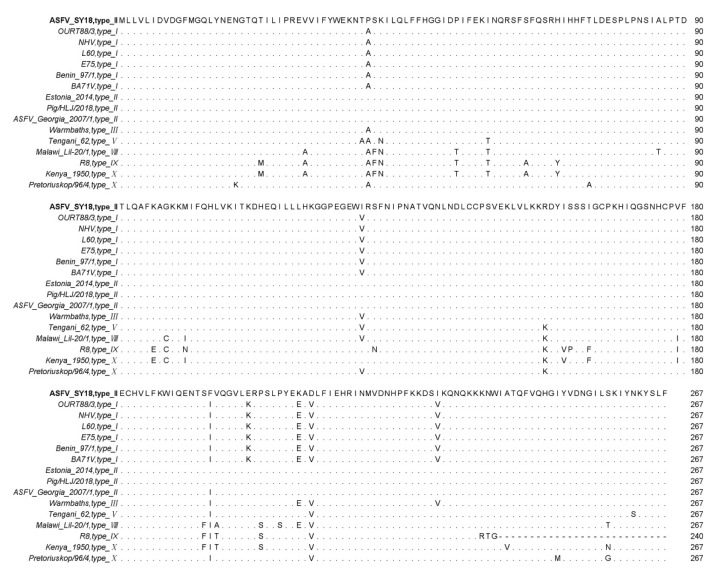
Multiple sequence alignment of ASFV pI267L amino acids residues. In multiple sequence alignment, the same amino acids are displayed by ‘.’, the absence of amino acids are displayed by ‘-’, and the differential amino acids are displayed by abbreviated letters of amino acids.

**Figure 2 viruses-14-00053-f002:**
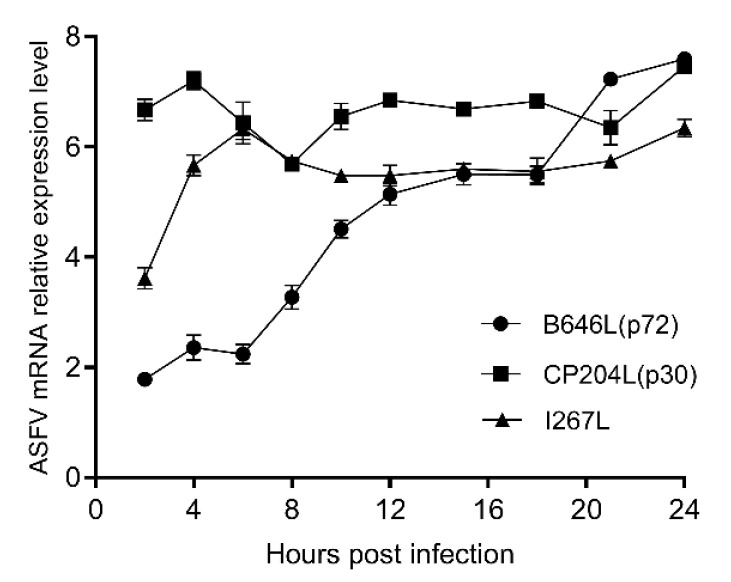
The relative expression levels of mRNA of *I267L CP204L* and *B646L*. The relative expression level of mRNA of *I267L*, *CP204L*, and *B646L* genes were quantified between BMDMs infected with SY18 and mock-infected BMDMs. The values of Y axis were expressed by the base-10 logarithm (log_10_) of the relative expression level.

**Figure 3 viruses-14-00053-f003:**
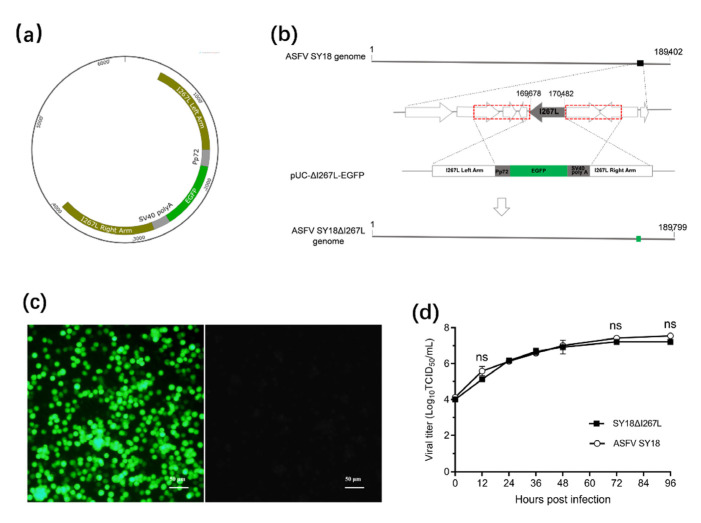
Construction of SY18∆I267L. (**a**) The recombinant plasmid, pUC-∆I267L-EGFP, was constructed. (**b**) Schematic representation of SY18ΔI226R construction. The location of the *I267L* gene was replaced with the EGFP cassette via homologous recombination between pUC-ΔI267L-EGFP and ASFV SY18 genomic DNA in vitro. The red dotted frame represents the position of the homology arms and the green square represents the position of EGFP. (**c**) BMDMs were infected with purified SY18ΔI267L and expressed green fluorescence (Left). The mock-infected BMDMs were non-fluorescence (Right). Bar 50 μm. (**d**) The viral titers of the two viruses were measured at 0, 12, 24, 36, 48, 72, and 96 hpi and exhibited using log_10_ TCID_50_/mL. They were non-significant differences (ns) at specific times (“ns” *p* ≥ 0.05).

**Figure 4 viruses-14-00053-f004:**
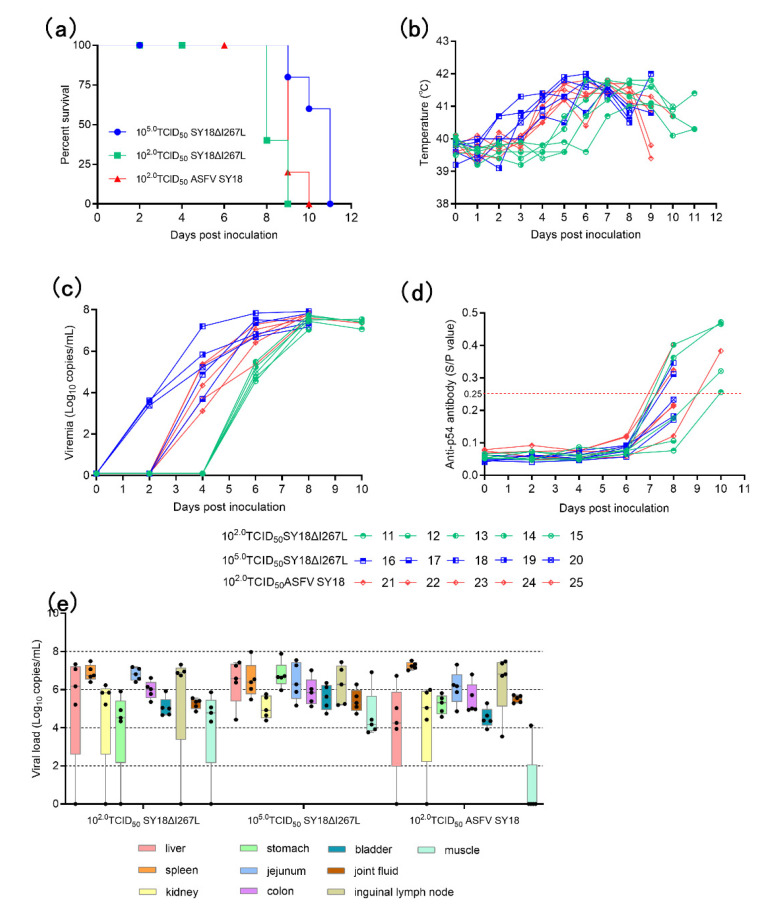
The results of survival rate, temperature, viremia, antibodies, and viral load of tissues. (**a**) The survival rate of the animal post inoculation with 10^2.0^ TCID_50_ and 10^5.0^ TCID_50_ SY18ΔI267L and 10^2.0^ TCID_50_ ASFV SY18. (**b**) The temperature of the animal post inoculation. (**c**) The viremia of the animal post inoculation. (**d**) The value of anti-p54 antibody of the animal post inoculation. The value of P (OD_450_)/N (OD_450_) greater than 0.25 is considered positive. The red dotted line represents P (OD_450_)/N (OD_450_) equal to 0.25. (**e**) The viral load in the tissues of the animal euthanized in extremis. The black dots represent the viral load of the individual in the tissue.

**Table 1 viruses-14-00053-t001:** Primers were used to assay gene expression by real-time quantitative PCR.

Gene	Forward Primer (5′–3′)	Reverse Primer (5′–3′)
*B646L*	CGAACTTGTGCCAATCTC	ACAATAACCACCACGATGA
*CP204L*	TTCTTCTTGAGCCTGATGTT	TAGCGGTAGAATTGTTACGA
*I267L*	GCCAATGCTTGAAGAGATG	ACCGTCCAGAACTTGAAC
*GAPDH*	CCTTCATTGACCTCCACTACA	GATGGCCTTTCCATTGATGAC

**Table 2 viruses-14-00053-t002:** Survival and fever responses of pigs inoculated with SY18ΔI267L and ASFV SY18.

Virus	No. of Survivors (*n* = 5)	Fever			Days of Viremia Onset (±SD)	Clinical Sympotoms	Mean Days to Death (±SD)
		Days of Onset (±SD)	Days of Duration (±SD)	Maximum Daily Temp.°C (±SD)			
ASFV SY18 (10^2.0^TCID_50_)	0/5	4.0 (±0)	5.4 (±1.14)	41.7 (±0.08)	6 (±0)	1. Fever (5/5) 2. Inappetence (4/5) 3. Diarrhea (4/5) 4. Arthrocele (3/5)	10.2 (±0.45)
SY18ΔI267L (10^2.0^TCID_50_)	0/5	6.0 (±0.71)	5.4 (±0.55)	41.7 (±0.18)	8 (±0)	1. Fever (5/5) 2. Inappetence (5/5) 3. Diarrhea (3/5) 4. Arthrocele (3/5)	11.4 (±0.89)
SY18ΔI267L (10^5.0^TCID_50_)	0/5	2.8 (±0.84)	6.4 (±1.14)	41.8 (±0.12)	5.2 (±1.10)	1. Fever (5/5) 2. Inappetence (4/5) 3. Diarrhea (3/5) 4. Arthrocele (2/5)	9.4 (±0.55)

Note: 1. Numero sign is abbreviated as No.. 2. Standard deviation is abbreviated as SD.

## Data Availability

Not applicable.

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
