# Peer review of "I267L Is Neither the Virulence- Nor the Replication-Related Gene of African Swine Fever Virus and Its Deletant Is an Ideal Fluorescent-Tagged Virulence Strain"

_viruses, 2021, doi:10.3390/v14010053_

Round 1

Reviewer 1 Report

The changes to the text the authors have made have improved the manuscript. However, I find it difficult to interpret the additional data with cytarabine.

Treatment of ASFV infected cells with cytosine-D-arabinofuranoside should block expression of late genes such as B646L, but not effect early genes such as CP204L. Therefore, unless I am missing something the authors data implies the opposite? Either way the authors new data suggests I267L is the same class of mRNA as CP204L, it is just the opposite of what readers would expect.

If I am correct, then the authors may have found something interesting, but it doesn't really help with the manuscript as it stands. My suggestion is to remove the additional data (really sorry!) and just state that I267L transcription is similar to CP204L without explicitly stating that I267L is an early gene. They could cite the Cackett paper or even the preprint in bioRxriv that suggests I267L is an early gene in infected macrophages. The rest of the authors data is solid and expands out knowledge about ASFV virulence and I don't think it should be held up over a relatively minor point.

Could the authors please clarify their mRNA expression level calculations in the methods. I assume the data is first normalised to GAPDH and then compared to mock?

Reviewer 2 Report

The study by Zhang et al. produced a mutant ASFV with I267L replaced with EGFP and further tested the mutant in pigs along with the parental virus. The design is more like a pathogenesis study of EGFP virus and parental virus. English language polishing is needed. My concerns are as below:

Major comments

  • The main claim of this study is the deletant is an ideal fluorescent-tagged virulence strain. However, besides growth properties, genetic stability of the EGFP virus as should be evaluated and can be done through sequencing GFP of serial passages in vitro. For stabilities in vivo, sequencing GFP from terminal tissues or blood should be conducted. Only mutant virus with good genetic stability can be utilized for further applications.
  • Line 153 of methods indicates survived pigs were challenged at 28 days. Please explain the rational for this design. Additionally, per table 2, mean days to death is around 10 days and if adding 3 SD, still true that around 15 days post inoculation, all pigs already died. Please provide more details of the performance of individual pigs throughout the study.
  • To prove the concept that I267L is not associated with virulence and replication. A deletant virus without replacement is needed.
  • The authors claim that I267L is an early gene based on relative expression of I267L, p30, p72 mRNAs. Figure 2b used inhibitors to treat cells which is a typical method to study viral infection stages. Please explain more details of using the cytarabine relation to early viral gene expression. Additionally, based on figure 2, p30/p72 are also expressed at a similar level as that of the I267L at early stages of infection. Structural proteins such as p30/p72 are supposed to be late genes as they are required to form new viral particles. Please explain your findings.

Minor comments

  • Line 42: “has been” should be “had been” and this sentence needs a reference citation.
  • Line 67: “a pool comprised the ASFV genes” should be “comprised of”
  • Line 71: “Induced” to “induce”.
  • Line 162: change “detected” to “tested”.
  • Line 258: “group” to “groups”.
  • Line 265: I believe it’s “copies/g”. please double check.
  • Line 288: “isolations” to “isolates”.
  • Line 289: “base mutations” to “nucleotide mutations”.

Round 2

Reviewer 2 Report

My concerns have been addressed.

This manuscript is a resubmission of an earlier submission. The following is a list of the peer review reports and author responses from that submission.